# Gender Differences in the Relationship between Physical Activity, Postural Characteristics and Non-Specific Low Back Pain in Young Adults

**DOI:** 10.3390/jfmk9040189

**Published:** 2024-10-09

**Authors:** Verner Marijančić, Stanislav Peharec, Gordana Starčević-Klasan, Tanja Grubić Kezele

**Affiliations:** 1Department of Physiotherapy, Faculty of Health Studies, University of Rijeka, 51000 Rijeka, Croatia; verner.marijancic@uniri.hr (V.M.); stanislav.peharec@fzsri.uniri.hr (S.P.); 2Department of Basic Medical Science, Faculty of Health Studies, University of Rijeka, 51000 Rijeka, Croatia; 3Department of Physiology, Immunology and Pathophysiology, Faculty of Medicine, University of Rijeka, 51000 Rijeka, Croatia; tanja.grubic@uniri.hr; 4Department of Clinical Microbiology, Clinical Hospital Rijeka, 51000 Rijeka, Croatia

**Keywords:** kyphosis, lordosis, low back pain, physical activity, posture, trunk muscle endurance, young adults

## Abstract

**Background/Aim:** University students are a particularly vulnerable population, as they spend increasing amounts of time sitting, which poses a major threat to their musculoskeletal health and posture. The aim of this cross-sectional study was to investigate gender differences in the relationships between physical activity (PA) and sedentary behavior, spinal curvatures and mobility, the endurance and balance of the trunk muscles, and the possible presence of non-specific low back pain (NS-LBP) in young adults aged 18–25 years. **Methods:** A total of 139 students completed all required tests. **Results:** Male students engaged in significantly more PA related to recreation, sports and leisure and were significantly more likely to be hyperkyphotic than female students. The more the male students participated in sports, the more pronounced the thoracic kyphosis. Female students had significantly more pronounced lumbar lordosis and anterior pelvic tilt that correlated with lumbar lordosis. Female students generally had significantly higher trunk extensor endurance and more balanced trunk musculature than males. NS-LBP correlated with PA in female students who generally had higher levels of NS-LBP than male students, with a statistically significant difference between those who practiced the most PA. **Conclusions:** Our results suggest that female students practice less PA and have pronounced lordosis and trunk extensor endurance, in contrast to males who practice more PA and have pronounced trunk flexor endurance and hyperkyphosis. Our findings suggest that more PA should be encouraged but implemented with caution and as an individualized gender-specific approach to prevent postural deformities and chronic musculoskeletal disorders, including NS-LBP.

## 1. Introduction

Low physical activity (PA) and sedentary behavior among young adults have become a major public health problem worldwide [1]. More than half of adolescents and a third of adults do not achieve the level of PA recommended by the World Health Organization (WHO) [2]. University students represent a particularly vulnerable population as they spend increasing amounts of time sitting during their studies, which poses a major threat to their musculoskeletal health and posture [3,4,5]. Healthy posture is defined as a state of balance between the muscles and spine that is essential for maintaining normal static and dynamic positions of the body [6].

The imbalance of the trunk extensor and trunk flexor muscles can alter the curvature of the spine, increase lordosis and kyphosis and cause health problems such as non-specific low back pain (NS-LBP) [7,8,9,10] and/or thoracic back pain, with low back pain being the main cause of disability [11,12,13]. In addition, excessive sitting and lack of PA are usually the causes of poor posture and a pronounced anterior pelvic tilt, which typically results from an imbalance of the muscles pulling on the pelvis and/or lumbar spine [14,15]. The increased anterior pelvic tilt also causes an exaggeration of the lumbar lordosis. Namely, the trunk extensors (e.g., *erector spinae*) are important for maintaining an upright posture and tend to become hypertonic when fatigued, exaggerating the lumbar lordosis and causing anterior pelvic tilt as part of the postural dysfunction [16,17]. The trunk flexors (e.g., *rectus abdominis*) tend to fatigue easily and become weakened when overstressed [18]. An increasingly sedentary time encourages overuse of the posture and trunk extensors at the expense of trunk flexors. Thus, the flexor muscles can become weak through disuse.

However, the angular degree of spinal curvatures depends not only on PA and a sedentary lifestyle but also on gender and the type and intensity of sport that young people engage in [19,20]. There have been found to be inherent differences between men and women in terms of hormonal factors, musculoskeletal structure and muscle strength, body composition and neuromuscular control that contribute to variations in posture [21,22]. The anatomical difference in musculoskeletal structure between the sexes may influence the distribution of body mass and alter the biomechanics of postural control [21,23,24]. In addition, hormonal factors such as estrogen and progesterone levels may affect ligament laxity and joint stability [24], and testosterone helps to improve maximal voluntary strength and power in men [25]. In addition, differences in sensory and motor strategies for neuromuscular control between the sexes may affect the ability to maintain stability and adapt to different postural demands [26]. Furthermore, PA influences ossification processes and muscle strength and is one of the most important factors influencing posture [19]. The most common postural abnormalities that occur in most sports are scoliosis and kyphosis, while lordosis occurs to a somewhat lesser extent [27]. The occurrence of such postural abnormalities in sports is usually related to the highly repetitive nature of the sports, certain specific exercises that put a lot of strain on the still underdeveloped spine, weakness of the muscular joints that can occur during puberty, etc. All of these factors can promote the occurrence of postural abnormalities and their further development [28,29]. In the relevant literature, there are many studies investigating PA and posture in healthy populations, including college students [30]. However, a recent meta-analysis showed no significant association between PA and human posture, although a weak correlation was found. The lack of a significant association may suggest that multiple biopsychosocial factors may be involved in human posture, as we have previously mentioned [30]. In contrast, other recent studies in young adults of both sexes have found a high percentage of postural abnormalities in individuals with low PA [31,32]. Despite an increasing number of studies focusing on university students, i.e., young adults aged 18–25 years, there is still a large gap in knowledge regarding the gender differences in the relationship between PA and posture, and the presence of back pain as a result of postural abnormalities.

Therefore, the aim of this cross-sectional study was to investigate the gender differences in the relationships between PA and spinal curvatures and mobility and the endurance and balance of the trunk muscles. Furthermore, in this study, we only focused on the presence of NS-LBP because the students in our pilot study only reported this pain as more pronounced, which is consistent with other studies that have examined students with sedentary habits [10,33].

## 2. Materials and Methods

### 2.1. Participants

The present study was a cross-sectional study conducted at the University of Rijeka, Croatia, during the academic school year 2022–2024. Participants were recruited from three different faculties: the Faculty of Health Studies, the Faculty of Medicine and the Faculty of Maritime Studies.

An estimation of the appropriate sample size for the study was guided by these two methods: (1) previous research and (2) general statistical principles; more specifically:(1)Based on earlier studies that used a similar methodology [34,35],(2)The program MedCalc (© 2023 MedCalc Software Ltd., Ostend, Belgium) estimated a minimum of 63 subjects which was needed to achieve 80% power with Cohen’s d = 0.50 for effect size, α = 0.05 type I error and beta = 0.20 type II error [36,37].

After the research project was introduced in a public presentation, an interview was conducted with 168 volunteers. After an interview, a total of 144 volunteers met the inclusion criteria and were enrolled in the study. The inclusion criteria were healthy university students aged 18 to 25 years, i.e., young adults, without cardiovascular, respiratory, metabolic, autoimmune, or other systemic diseases and/or spinal pathologies, without a previous diagnosis of a systemic musculoskeletal problem or pain and without a history of spinal or limb surgery. Individuals who used assistive devices or orthoses were also excluded from the study.

Of these 144, a total of 139 completed all required tests. The researchers obtained the necessary approvals from the Ethics Committee of the Teaching Institute of Public Health (number: 08-820-40/50-22) and the Ethics Committee of the Faculty of Health Studies of the University of Rijeka (number: 2170-1-65-23-1). Informed consent was obtained from all participants in accordance with the Declaration of Helsinki.

### 2.2. Study Design

The design of the cross-sectional study followed the STROBE Statement [38]. All measurements were carried out by the same researchers. A research protocol designed for this study consisted of three phases. The 1st phase consisted of completing the questionnaires as previously described, including the International Physical Activity—Long Form (IPAQ-LF) [32]. The 2nd phase consisted of 2 subphases, as previously described [32], where trunk muscle endurance tests were performed. The 3rd (final) phase included the measurement of spinal curvatures. There was a 24 h break between the three main phases.

### 2.3. Outcome Measures

#### 2.3.1. Self-Reported PA and Time Spent Sitting

The IPAQ-LF was administered by trained interviewers to assess participants’ self-reported PA and sedentary behavior [39,40]. It is a reliable and valid questionnaire that health education and promotion professionals can confidently use to assess college students’ participation in PA [41]. The validity indices of the questionnaire are similar to other self-reported PA questionnaires [42]. A detailed description of the IPAQ scoring protocol, including the criteria for cutting off extreme values, is available online [39].

#### 2.3.2. Visual Analogue Scale (VAS) for NS-LBP

The VAS for pain was used to assess the intensity of NS-LBP in the last 4 weeks [43,44]. This is a valid and reliable test for measuring subjective characteristics or attitudes that cannot be measured directly. Here, participants indicated their level of pain by selecting the appropriate number under the picture of the facial expression and the corresponding description. The VAS for physical pain was 10 units long (0 = no pain and 10 = worst possible pain).

#### 2.3.3. Trunk Muscles Endurance Testing

##### Trunk Extensor Endurance Testing

The trunk extensor endurance test is a reliable and valid test for assessing the muscular endurance of the torso extensor muscles that stabilize the spine (i.e., *erector spinae* and *multifidus* muscles) [45,46,47]. It is a timed test with a static, isometric contraction performed according to the modification according to McGill et al. [45]. Participants were instructed to lie on a test table in a prone position. The trunk was positioned at the level of the anterior superior iliac spine at the edge of the test table (Figure 1A). Participants kept their upper body away from the end of the table by supporting themselves with their outstretched arms on a chair directly below them. The test time was set at 180 s and measured with a stopwatch while the arms were lifted from the chair and crossed over the chest, with the hands resting on the opposite shoulders and the participants assuming the horizontal position (Figure 1B). Researcher 1 stood by the side and measured the time, and the test was terminated when participants deviated from the horizontal plane. Researcher 2 stabilized the participants’ lower body by holding the participants’ lower extremities down [47].

##### Trunk Flexor Endurance Testing

A standardized trunk flexor endurance test was performed according to previously published methods [45]. The trunk flexor endurance test is a reliable and valid test that assesses the muscular endurance of the trunk flexors (i.e., *rectus abdominis*, external obliques, internal obliques and *transversus abdominis* muscles) [46,47]. This is a timed test in which the anterior muscles are isometrically contracted to stabilize the spine until the subject shows signs of fatigue and can no longer maintain the assumed position or reach the predetermined time of 180 s. The test was performed in a supine position. Participants were in a supine position with the hips and knees flexed to 90° and the trunk resting on a wedge at a 60° angle (Figure 2A). The arms were crossed in front of the chest and the hands were placed on the opposite shoulders. Time was measured from the moment the wedge was pushed back 10 cm until the participant re-established contact with the wedge (Figure 2B). Researcher 1 stood at the participant’s side and measured the time with a stopwatch. Stabilization of the participant’s feet was performed by the researcher 2 [47].

##### Balance of Trunk Muscles

The trunk extensor/flexor endurance test ratio represents a good parameter for the balance of the trunk musculature. It is a ratio between the endurance of the trunk extensors and the endurance of the trunk flexors. This measure is calculated from the ratio between the trunk extensor endurance and the trunk flexor endurance scores. There are no reference values for this ratio. It was modified following Kim et al. [48].

#### 2.3.4. Evaluation of Spinal Curvatures

Spinal curvatures (angle of thoracic kyphosis and lumbar lordosis, and sacrum–hip angle) were measured using a non-invasive Spinal Mouse^®^ (SM) device (Idiag M360, Fehraltorf, Switzerland). This is a safe, reliable, quick and easy-to-use method with no side effects and a suitable substitute for X-rays to measure spinal and pelvic alignment and mobility including kyphosis, lordosis and pelvic tilt [49,50,51,52,53,54,55]. It is a skin-surface device that can be used in different body positions, i.e., upright standing and forward bending. The SM has acceptable metrological properties for assessing sagittal thoracic and lumbar curvature and spinal mobility. Its intraclass correlation coefficients (ICCs) for intrarater reliability are between 0.61 and 0.96 and the ICCs for interrater reliability are between 0.70 and 0.93 [50,51,52,53,54,55].

The thoracic kyphosis and lumbar lordosis were measured in standing and flexion. The pelvic tilt was measured in the standing position. The postural measurements were performed in the sagittal plane with bare feet in a relaxed standing position, i.e., in anatomical position using the Idiag M360 protocol software version G6 6.4 2X. The measurements were performed in one day, and no exercises were performed before the measurement. Using the software of this device, the data displayed on the screen were used to analyze the positional relationship between each vertebra, measure the angles between the vertebrae and calculate the angles of the spinal curvatures. The standard procedure for the upright sagittal posture was performed; the spinous process of the 7th cervical vertebra was marked as the starting point for the measurement and the end point was marked at the level of the 3rd sacral vertebra. The posterior superior iliac spine (PSIS) was marked using an alternative method by drawing the line between the PSIS. After the line between PSIS was drawn 2 cm below the line, the position was marked with a flexible ruler. The vertical line was used to mark the center of the new line below the PSIS line so that the cross was over the S3 vertebra. The SM is placed over the C7 vertebra with the orange mark on the device over the marked starting position and the recording is made by moving the device from top to bottom to the end point. In the flexed posture, the upper body is bent as far as possible and the arms and head hang freely. The knees and legs are straight. Before starting the measurements, the starting position marked on the skin is moved upwards by 2 cm. The values of the thoracic spine (T1/T2 to T11/12) and the lumbar spine (T12/L1 and L1 to S1), as well as the sacrum–hip angle, were recorded. Negative values in the lumbar curve correspond to lumbar lordosis. When assessing the thoracic spine in a standing position, values between 20° and 45° were considered neutral thoracic kyphosis, less than 20° was considered hypokyphosis and more than 45° was considered hyperkyphosis [49]. The values of the lumbar spine were considered for a neutral lordosis if ranging from 20° to 40°, below 20° were classified as hypolordosis and more than 40° as hyperlordosis [56]. The values of pelvic tilt were considered neutral pelvic tilt if they were between 10° and 15°, below 10° they were considered posterior pelvic tilt and more than 15° anterior pelvic tilt [57].

### 2.4. Statistical Analysis

The data were analyzed using *Statistica*, version 13 (TIBCO Software Inc., 2017, Palo Alto, CA, USA). The genders (independent variables) were compared by descriptive data and different domains of the IPAQ-LF: age, body mass index (BMI) (kg/m^2^), total PA (MET-min/wk), the most frequently performed type of PA (recreation, sport and leisure in MET-min/wk) and by time spent sitting (h/day).

As there are no established thresholds for presenting MET-minutes (dependent variables), the IPAQ Research Committee proposes that these data be presented as comparisons of median values and interquartile ranges for different populations [25,58]. When calculating the energy expenditure in MET-min/wk, all individuals who exceeded 30,561 MET-min/wk were marked as outliers and excluded from the study in accordance with the instructions for calculating the IPAQ questionnaire.

Therefore, participants were categorized into quartiles of PA levels using the IPAQ-LF: 672–2924, 2925–4759, 4760–7989 and 7990–30,561 MET-min/wk. Accordingly, dependent variables (percentage of each gender, angle for thoracic kyphosis and lumbar lordosis and angle for sacrum–hip (pelvic tilt) in standing position and flexion, trunk muscle endurance (trunk flexor and extensor endurance in sec), ratio of extensors/flexors of the trunk (balance), and VAS for NS-LBP) were compared between the quartiles of the same PA level of both genders.

Throughout the text, the following symbols were used for PA quartiles: “Q1”, “Q2”, “Q3” and “Q4”. The data distribution was tested for normality using the Kolmogorov–Smirnov test. Gender was presented as a percentage, age, BMI, PA, time spent sitting, spine angles and muscle endurance as mean ± SD. The VAS for NS-LBP was presented as median and range. To compare dependent parametric variables (age, BMI, PA, sitting time, spine angles and muscle endurance) between genders, we used the Student *t*-test, and to compare the dependent non-parametric variable VAS, we used the Mann–Whitney U test.

Chi-square analyses were used to examine the frequency distributions of the genders between the PA quartiles.

The relationships between lordosis, boot extensor endurance, pelvic tilt, extensor-flexor balance and VAS for NS-LBP, time spent sitting and total PA per week were analyzed using Pearson correlation. In the correlation analyses, the values of the correlation coefficients were considered as follows: 0.00–0.19 was considered as “no relationship”, 0.20–0.39 as “weak relationship”, 0.40–0.69 as “medium relationship”, 0.70–0.89 as “strong relationship” and 0.90–1.00 as “very strong relationship”. As the *p*-values alone do not give any indication of the size of an effect, we calculated the effect sizes for the differences between the genders as Cohen’s d and interpreted them as criteria: small (0.2), moderate (0.5) and large (0.8) [37]. The significance level of the statistical analyses was set at *p* < 0.05.

## 3. Results

At baseline, a total of 168 subjects were enrolled in the study, and after the exclusion of 24 subjects who had not completed all the required assessments, the study was completed with 139 subjects (Figure 3). Of these, 82 (59%) were female and 57 (41%) were male.

No significant differences in BMI and age were found (Table 1). The BMI values reflect the normal weight of these young adults (23.5 ± 2.7). The mean age of the participants was 21.0 ± 2.0 years, and the mean BMI was 23.5 ± 2.7 kg/m^2^. The descriptive data and the different domains of the IPAQ questionnaire for all 139 subjects can be found in Table 1. The mean ± SD of total PA was 5379.2 ± 4911.4 for female students and 7531.0 ± 5152.7 MET-min/wk for male students, with a statistically significant difference (*p* = 0.023). Overall, male students practiced more PA than female. The type of PA with the highest number of MET-min/wk is PA during recreation, leisure and sport (mean ± SD = 2837.7 ± 2458.0 MET-min/wk), and it was shown that male students practiced this type of PA significantly more often (3471.3 ± 2673.9 vs. 2387.8 ± 2203.2, *p* = 0.017, *d* = 0.44). Although female students practiced more PA in relation to domestic activities and at work, there were no statistically significant differences [59]. The time spent sitting was similar for both genders (5.9 h per day for females vs. 5.4 h per day for males), with no significant difference and small effect size (*d* = 0.2).

Additionally, statistical analyses revealed statistically significant differences in the percentage of participants practicing PA between PA quartiles (*p* = 0.033), i.e., the quartile with the highest PA (Q4) had significantly more male participants (57.2% vs. 42.8%) and the quartile with the lowest PA (Q1) had more female participants (79.4% vs. 20.6%) (Table 2).

Overall, standing measurements revealed 46% of hyperkyphotic students, of which 52% were male, and most students had neutral kyphosis (54%) with a majority of female students (67%). There were no hypokyphotic students. The mean score of thoracic kyphosis in standing was higher or hyperkyphotic in male students than in female students (46.3 ± 9.4 vs. 42.6 ± 9.7, *p* = 0.040, *d* = 0.38), with statistical significance occurring in the last two quartiles with the highest PA (Q3 and Q4) (Table 2). It appears that the male students with more exercise have a more pronounced thoracic kyphosis. In the flexed posture, hyperkyphosis was even more pronounced in both genders in all PA quartiles.

Overall, 18% hyperlordotic students were found, of whom 91% were female; 9% students were hypolordotic, of whom 80% were male and 73% had neutral lordosis, of whom 57% were female. The mean value of lumbar lordosis in standing was significantly more negative in female students than in males (−34.3 ± 8.9 vs. −26.3 ± 7.9, *p* < 0.001, *d* = 0.95) with similar values in all PA quartiles (Table 2). In flexion, however, the lordosis angle changed in the opposite direction for both genders, i.e., the male students had a significantly greater lordosis angle than the female students (−35.1 ± 8.4 vs. −25.8 ± 9.1, *p* < 0.001, *d* = 1.06) (Table 2).

We can actually determine the mobility of the spine based on the length of the path that the spine takes from the standing position to the final position in flexion. Therefore, if female students have a higher lumbar lordosis angle while standing, they will achieve a lower flexion score because they have to overcome the angle from lumbar lordosis to neutral position and then from neutral position to maximum flexion in order to achieve a certain degree of lordosis. Male students have a shorter path, so they achieve higher results in flexion. Therefore, it can be said that both genders have good flexibility in the lumbar spine (Table 2).

Overall, 47% of the students were found to have anterior pelvic tilt while standing, of whom 93% were female; 19% of the students had posterior pelvic tilt, of whom 82% were male and 34% of the students had neutral pelvic tilt, of whom 68% were male. The mean value for pelvic tilt was significantly higher in the female students, indicating pronounced anterior pelvic tilt, in contrast to the male students who had neutral pelvic tilt (19.2 ± 6.2 vs. 11.0 ± 4.0, *p* < 0.001, *d* = 1.57) (Table 2), which was strongly correlated with lumbar lordosis in both genders (f: *r* = 0.838 and m: *r* = 0.746) (Table 3). However, this correlation was stronger in female students. In addition, a weak but statistically significant correlation between pelvic tilt and trunk extensors (*r* = 0.290) was found only in male students (Table 3). The stronger the trunk extensors, the greater the anterior pelvic tilt in males. Female students showed higher values for trunk flexor endurance than male students did but without a statistically significant difference (170.0 ± 26.1 vs. 168.9 ± 27.4, *p* = 0.827, *d* = 0.04). In addition, trunk extensor endurance values in Q1 with the lowest PA were higher in both genders unlike those values in Q4 with the highest PA. However, female students generally showed significantly higher values of trunk extensor endurance than male students did (154.3 ± 34.5 vs. 140.5 ± 37.9, *p* = 0.045, *d* = 0.38), which is related to a more pronounced lumbar lordosis (Table 2).

According to the ratio of trunk extensor/flexor endurance (balance between trunk ex-tensors and flexors), both genders had higher endurance of trunk extensors than flexors. However, female students again had significantly higher ratio values and accordingly more balanced trunk muscles (0.92 ± 0.2 vs. 0.81 ± 0.2, *p* = 0.019, *d* = 0.55) (Table 2). The students who practiced PA the least (Q1) had the most imbalances in their trunk muscles (f: 0.88 ± 0.1 and m: 0.74 ± 0.2, *p* = 0.042, *d* = 0.88). The balance between extensors and flexors was weakly but significantly correlated with lordosis angle in male students (r = 0.363) and moderately correlated with trunk extensor endurance in female students (r = 0.625) (Table 3). The greater the lordosis, the more unbalanced the relationship between the trunk muscles in males, and the greater the lordosis, the greater the endurance of the trunk extensors in females.

Of the participants who experienced NS-LBP (69%), 59% were female students. The median VAS score of those reporting NS-LBP was 2 with a range of 1 to 8. Female students generally showed a higher level of NS-LBP than male students did with a statistically significant difference in Q4 with the highest PA level (*p* = 0.040) (Table 2). In addition, we found a weak but significant correlation between PA and NS-LBP (Table 4), suggesting that the female students who practiced more PA had higher levels of NS-LBP. However, we found no correlation between the VAS (f: *r* = 0.003 and m: *r* = 0.157) and lordosis angle or trunk extensor endurance (f: *r* = −0.082 and m: *r* = 0.123) in either gender (Table 3), nor between the VAS and time spent sitting (Table 4).

## 4. Discussion

Our study has shown that female students have a significantly more pronounced lumbar lordosis with a more pronounced anterior pelvic tilt than usual and greater strength of the lower back muscles (trunk extensors), which indicates a postural problem, although the trunk muscles are more balanced. In addition, we found a correlation between NS-LBP and PA in females. On the other hand, male students have more pronounced endurance of trunk flexors and more pronounced hyperkyphosis, especially those who practice more PA. No correlation was found between NS-LBP and PA in males.

### 4.1. PA and Time Spent Sitting in Young Adults

According to the priorities of the WHO’s GAPPA 2018–2030 [2], young adults should practice more PA and reduce sedentary behavior. In addition, it was shown that young men are more physically active because they have a greater interest in practicing sports, especially high-intensity sports, whereas girls tend to engage in more individual and moderate-intensity exercise and favor more sedentary activities [60]. In our study, male students performed more PA during one week, especially PA related to recreation, sports and leisure. From our previous analyses [59], female students practiced more PA related to household and work. The research found that computer use in male students was negatively correlated with time spent exercising [61]. In contrast, in female students, TV watching was negatively correlated with vigorous physical activity [60]. Our previous results have also shown that male students statistically engage in more vigorous PA than females [59]. In addition, female students in our study spend more time sitting, but without statistically significant differences. Research has found that sitting has a negative impact on health when it exceeds 7.5 h per day [62,63].

In addition, self-reported data showed that university students spend 7.3 h/day sitting, but the level of sedentary behavior was significantly higher when measured with accelerometers (mean = 9.8 h/day) [64]. This suggests that the actual time spent in sedentary behavior may be higher than the 5.9 h (female students) and 5.4 h (male students) measured in our study.

### 4.2. Posture and NS-LBP in Young Adults

As in our previous study, we did not find a significant association between spinal curvatures with a certain kind of intensity, type of PA or time spent sitting [32]. Similar results were shown in a recent study by Grabara et al., 2024 as well, although the methods for measuring spinal curvatures differed [65]. However, we have found posture differences between genders. Male students exhibited more pronounced hypekyphosis in standing position and less deep lumbar lordosis. On the other hand, female students exhibited more pronounced lordosis and anterior pelvic tilt without pronounced hyperkyphosis. As already said, sedentary behavior and lack of PA can cause the muscles supporting the spinal curvatures to weaken or on the other hand to become hypertonic [16,17,18]. These changes can lead to postural abnormalities as the body tries to maintain correct alignment without the necessary muscular support. Although we did not find a significant difference in sedentary behavior between genders, female students did spend more time sitting and practiced less PA, which could be connected to postural changes we have found in them. One of these abnormalities is lumbar lordosis [32]. In addition, certain types of PA or sport can affect spinal curvatures, due to the specific positions and/or movement patterns performed during exercise, with repetitive performances that can cause different adaptations and therefore postural abnormalities [66,67,68]. These effects depend on the type of sport and training intensity and can have a more or less favorable effect on posture. Other specific forms of PA or exercise programs such as yoga, Pilates and dance can positively influence posture [69,70,71] by improving core strength, which is crucial for maintaining correct proper alignment. Moreover, the systematic review from González-Gálvez et al., 2019 suggests that strengthening rather than stretching could be more relevant for kyphosis and both qualities are important for lordosis [72]. However, some sports at an early age can negatively affect the posture that can be maintained in future life [28]. Also, involving children in the training process at a very early period in their childhood when the spine is affected by the influence of large loads can lead to adaptive changes in the musculoskeletal system and disrupt normal posturogenesis [73]. Kyphosis as a postural abnormality more often occurs in most sports, while lordosis occurs to a somewhat lesser extent [27]. Is this a reason why male students who practice more PA in our study have pronounced kyphosis, it should be further explored. In addition, the male students in our study have more pronounced endurance of trunk flexors than extensors indicating that exercise engagement for strengthening abdominal muscles by males could additionally cause kyphosis because of their thickening and shortening [74,75]. On the other hand, the reason why the female students exhibit more pronounced lordosis and anterior pelvic tilt than males could be associated with the sex-specific loading demands (e.g., pregnancy loads) to accommodate increased upper body load and an anteriorly displaced center of mass during future pregnancy [76].

NS-LBP rates among younger individuals are rising. Underlying changes in posture and trunk behavior may be responsible for its occurrence [77]. It has been linked to various conditions including obesity, increased lumbar lordosis, low abdominal muscle strength, imbalance between flexor and extensor trunk muscle strength, reduced spinal mobility, etc. [78]. Thus, it could be related to pronounced lordosis [79] and anterior pelvic tilt [80,81,82], which we found in female students, or a more pronounced imbalance in the trunk muscles in males. In addition, an imbalance in trunk muscle strength that we found in both genders can significantly influence the lordosis curve of the lumbar spine and might be one risk factor for potential lower back pain. In our study, both genders reported suffering from NS-LBP, especially men even up to level 8/strong pain according to VAS, although we found no correlation between pain, trunk muscle balance, lordosis, pelvic tilt, endurance of the trunk extensors, nor PA or sitting duration. Similar results with no direct correlation between NS-LBP and biomechanical changes in posture were found in the study by Marinho and Lucena 2022, although the majority of participants experienced NS-LBP [83]. The reason for the non-significant results could be the small number of participants in that study. On the other hand, other studies have found an association between NS-LBP and sedentary behavior as well as lumbar spine misalignment and lumbar lordosis [9,10,84]. In addition, some studies showed an association between female gender and NS-LBP [9,10,85] and some studies showed no association between NS-LBP and gender [86]. However, we found a difference between genders in Q4 with the highest level of PA, where female students reported a higher level of NS-LBP. Additionally, as confirmed in our previous analyses [59], female students practiced more PA related to the household and work. At this moment, it is not sure whether female students experience this pain because of the extent of PA or because of the wrong body positions practiced during the training, or some other PA that has a negative side effect on their body and posture. This should be in more detail investigated. As well, in our study, female students also spent more time sitting but without a significant correlation to NS-LBP. We assume that more objective measurements, such as measuring sitting time with accelerometers, would show a more realistic association. We also know that sport was reported as a risk factor for developing NS-LBP in more than half of studies examining it, especially in those studies, which assessed activities implying high or repeated loading on the spine [87]. Back pain is common in professional athletes, with an estimated prevalence ranging from 1% to 30%, and the prevalence of NS-LBP in recreational athletes is not known [88]. Even for children aged 10–12, NS-LBP is present depending on engaging in certain PA. The highest incidence of NS-LBP was detected among children who practiced volleyball, gymnastics and swimming for over 4 h a week, as well as among those who practiced rhythmic gymnastics.

Based on the results obtained, we can assume that NS-LBP could be affecting the young population, especially females, and it is possible that the type of sport as well as the frequency of exercise may determine whether a particular sport could be a risk factor associated with NS-LBP [89]. However, these findings should be further confirmed and investigated in more detail.

## 5. Strengths and Limitations

The study has strengths and limitations.

The strengths are as follows: (1) The greatest strength is the complexity of the assessments with various questionnaires and physical tests. (2) We used the most up-to-date recommended methods for this type of study. (3) We assessed PA using the long version of the IPAQ, dividing the results (in MET-min/wk) into PA quartiles (levels of PA) and comparing them between the genders according to dependent variables (spinal curvatures, the endurance and the balance of the trunk muscles and VAS for NS-LBP). (4) We used a non-invasive device, namely, the Spinal Mouse^®^, a safe, reliable, quick and easy-to-use method without side effects and a suitable substitute for X-rays to measure the angular values of the spine.

The limitations are as follows: (1) This includes its cross-sectional type of study. Therefore, statements regarding cause and effect cannot be made. (2) One of the main limitations is the complexity of the study due to the multiple measurements, which require participants to make extra effort to complete all tests. (3) The college students were predominantly from the Faculty of Health Studies, Medicine, and Maritime Studies, and we did not include young adults who do not study. (4) The study included more female students. (5) In addition, we did not analyze thoracic spine pain. (6) The study also lacked an objective measurement of PA and sedentary behavior. (7) Pain measurement should include more precise descriptions such as Brief Pain Inventory. Additional prospective studies using the objective measures of PA and sitting time are needed to confirm the findings of this study. (8) The device SM has its limitations, as it does not measure the cervical part of the spine. (9) Because of the small number of male participants in Q1, the validity of the statistical comparisons in this group could be low.

## 6. Conclusions

Our study has shown that female students have a more pronounced lumbar lordosis with a more pronounced anterior pelvic tilt than usual and more pronounced lower back muscle strength (trunk extensors), indicating a postural problem. We also found a weak but significant correlation between PA and NS-LBP in females. These postural differences could be one of the reasons why the female students had higher levels of NS-LBP when they practiced more PA. However, the question of how NS-LBP is related to postural abnormalities in females should be further investigated.

On the other hand, male students have pronounced endurance of trunk flexors and more pronounced hyperkyphosis, especially those who practice more PA.

More PA has considerable health benefits but can also have a negative impact on posture and musculoskeletal health if overused or incorrectly trained. Thus, the type of PA and the way the young adults perform their weekly training should not be ignored. This suggests that more PA needs to be encouraged but implemented with caution and as an individualized gender-specific approach to prevent postural abnormalities and chronic musculoskeletal disorders, including NS-LBP, especially because men and women differ in terms of structure, muscle strength and hormonal and psychophysical status.

## Figures and Tables

**Figure 1 jfmk-09-00189-f001:**
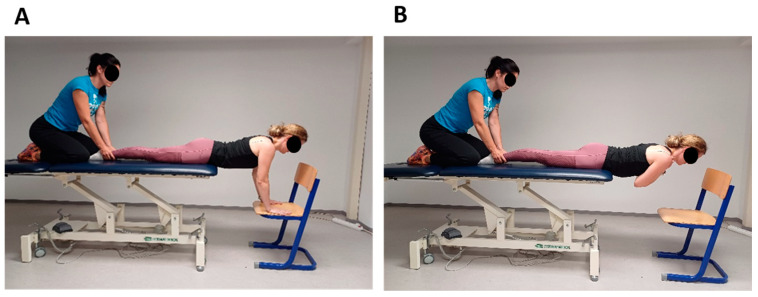
Trunk extensor endurance testing. (**A**) Participants held their upper body away from the end of the table by leaning on a chair directly below them with their arms outstretched. (**B**) The test time was set at 180 s and measured with a stopwatch while the arms were raised from the chair and crossed over the chest with the hands resting on the opposite shoulders and the participants assuming the horizontal position.

**Figure 2 jfmk-09-00189-f002:**
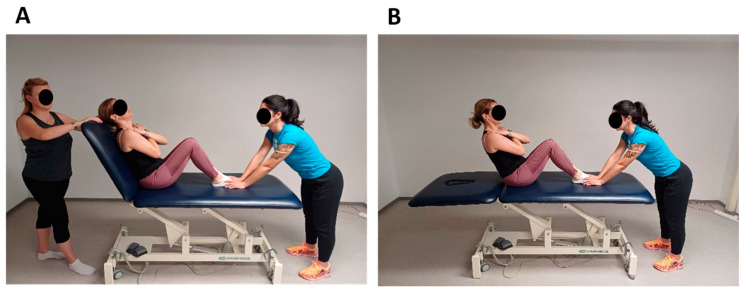
Trunk flexor endurance testing. (**A**) Participants were in a supine position with the hips and knees flexed to 90° and the trunk resting on a wedge at a 60° angle. (**B**) The time was measured from the moment the wedge was pushed back 10 cm until the participant reestablished contact with the wedge again.

**Figure 3 jfmk-09-00189-f003:**
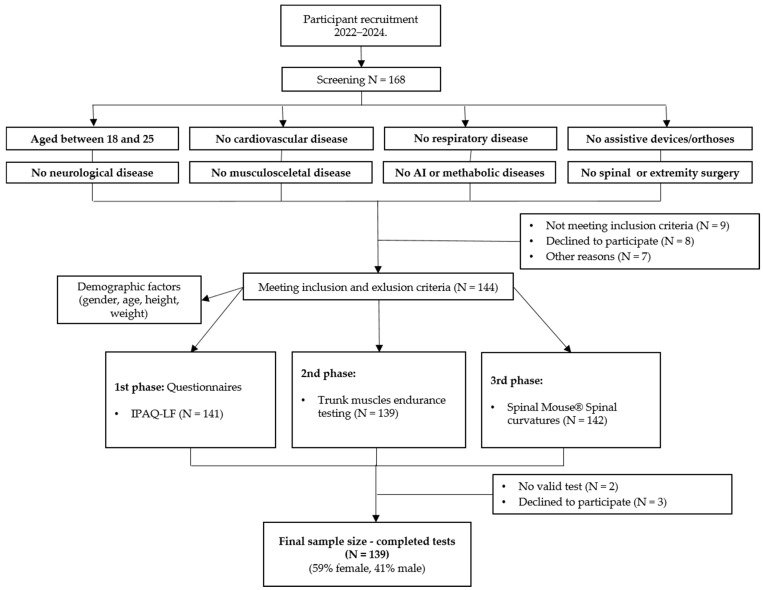
Flowchart of a number of participants at different stages in the study. AI, autoimmune; N, number; IPAQ-LF, the International Physical Activity Questionnaire—Long Form.

**Table 1 jfmk-09-00189-t001:** Demographic characteristics and gender differences in different domains of the IPAQ-LF (N = 139).

Variable	Females	Males	Total	*p* Value	*d*
Gender, n (%)	82 (59)	57 (41)	139 (100)	NA	NA
Age, mean ± SD	20.8 ± 1.8	21.3 ± 2.1	21.0 ± 2.0	0.151	0.25
BMI (kg/m^2^), mean ± SD	23.5 ± 2.8	23.6 ± 2.6	23.5 ± 2.7	0.861	0.03
Total PA (MET—min/wk), mean ± SD	5379.2 ± 4911.4	7531.0 ± 5152.7	6005.6 ± 4821.8	0.023 *	0.42
Recreation, sport and leisure (MET—min/wk), mean ± SD	2387.8 ± 2203.2	3471.3 ± 2673.9	2837.7 ± 2458.0	0.017 *	0.44
Sitting (h/day), mean ± SD	5.9 ± 2.0	5.4 ± 2.8	5.7 ± 2.3	0.301	0.20

NA, not applicable; n, number; SD, standard deviation; BMI, body mass index; PA, physical activity; MET, metabolic equivalent of task; wk, week; h, hour. Statistical analysis: Student *t*-test; * statistical significance, *d* Cohen’s value for effect size.

**Table 2 jfmk-09-00189-t002:** Comparison of variables between genders according to the quartiles of self-reported PA.

Variable (per PA Quartiles/MET-min/wk)	Females	Males	Total	*p* Value	*d*
Q1 (672–2924)					
Q2 (2925–4759)					
Q3 (4760–7989)					
Q4 (7990–30,561)					
^a^ Number of participants (%)				(*df* = 3) 0.033 *	NA
Q1	27/34 (79.4)	7/34 (20.6)	34/139 (24.5)		
Q2	24/36 (66.7)	12/36 (33.3)	36/139 (25.8)		
Q3	16/34 (47.1)	18/34 (52.9)	34/139 (24.5)		
Q4	15/35 (42.8)	20/35 (57.2)	35/139 (25.2)		
^b^ Kyphosis straight standing (°), mean ± SD					
Q1	44.2 ± 12.2	44.3 ± 10.3	44.1 ± 9.7	0.977	0.01
Q2	40.8 ± 9.6	44.1 ± 7.2		0.354	0.38
Q3	44.9 ± 7.5	44.7 ± 7.5		0.938	0.02
Q4	40.1 ± 7.3	50.0 ± 11.0		0.001 *	1.06
Total	42.6 ± 9.7	46.3 ± 9.4		0.040 *	0.38
^b^ Kyphosis flexion (°), mean ± SD					
Q1	53.8 ± 10.6	62.8 ± 5.6	58.6 ± 11.4	0.032 *	1.06
Q2	55.5 ± 13.7	61.7 ± 8.6		0.207	0.54
Q3	55.3 ± 12.5	65.0 ± 7.6		0.018 *	0.93
Q4	60.1 ± 11.8	61.2 ± 11.6		0.804	0.09
Total	55.8 ± 12.1	62.6 ± 9.0		0.001 *	0.63
^b^ Lordosis straight standing (°), mean ± SD					
Q1	−34.6 ± 8.7	−23.8 ± 9.9	−30.9 ± 9.4	0.008 *	1.15
Q2	−31.7 ± 8.5	−26.7 ± 5.7		0.105	0.69
Q3	−36.7 ± 9.3	−26.3 ± 7.4		0.002 *	1.23
Q4	−34.9 ± 9.6	−27.2 ± 8.7		0.030 *	0.84
Total	−34.3 ± 8.9	−26.3 ± 7.9		<0.001 *	0.95
^b^ Lordosis flexion (°), mean ± SD					
Q1	−28.6 ± 8.9	−33.8 ± 11.4	−29.7 ± 9.9	0.206	0.50
Q2	−27.9 ± 8.8	−36.7 ± 6.4		0.009 *	1.14
Q3	−21.1 ± 7.5	−34.0 ± 7.0		<0.001 *	1.77
Q4	−23.6 ± 9.8	−35.6 ± 9.4		0.002 *	1.24
Total	−25.8 ± 9.1	−35.1 ± 8.4		<0.001 *	1.06
^b^ Pelvic tilt (sacrum–hip angle) (°), mean ± SD					
Q1	19.0 ± 5.9	11.1 ± 4.8	15.8 ± 6.7	0.002 *	1.46
Q2	17.8 ± 7.4	10.9 ± 3.6		0.010 *	1.18
Q3	19.5 ± 5.6	11.0 ± 3.6		<0.001 *	1.80
Q4	21.4 ± 5.5	11.1 ± 4.4		<0.001 *	2.06
Total	19.2 ± 6.2	11.0 ± 4.0		<0.001 *	1.57
^b^ Trunk flexor endurance, mean ± SD					
Q1	169.8 ± 26.8	166.1 ± 27.5	174 ± 75.0	0.742	0.10
Q2	172.0 ± 21.9	180.0 ± 0.1		0.265	0.51
Q3	170.7 ± 24.8	157.6 ± 43.4		0.322	0.37
Q4	166.5 ± 34.0	164.6 ± 31.1		0.875	0.05
Total	170.0 ± 26.1	168.9 ± 27.4		0.827	0.04
^b^ Trunk extensor endurance, mean ± SD					
Q1	158.3 ± 34.2	156.6 ± 38.5	147.0 ± 37.3	0.905	0.04
Q2	151.6 ± 30.3	133.8 ± 37.8		0.172	0.51
Q3	161.6 ± 30.3	136.7 ± 26.8		0.027 *	0.87
Q4	143.3 ± 45.2	135.4 ± 46.6		0.641	0.17
Total	154.3 ± 34.5	140.5 ± 37.9		0.045 *	0.38
^b^ Extensors/flexors ratio (balance), mean ± SD					
Q1	0.88 ± 0.1	0.74 ± 0.2	0.91 ± 0.4	0.042 *	0.88
Q2	0.89 ± 0.3	0.84 ± 0.3		0.696	0.16
Q3	0.96 ± 0.3	0.93 ± 0.1		0.789	0.13
Q4	0.95 ± 0.1	1.03 ± 0.6		0.647	0.18
Total	0.92 ± 0.2	0.81 ± 0.2		0.019 *	0.55
^c^ VAS, median (range)					
Q1	1 (0–6)	2 (0–8)	2 (0–8)	1.000	NA
Q2	2 (0–6)	1 (0–6)		0.355	
Q3	1 (0–7)	2 (0–5)		1.000	
Q4	2 (3–7)	2 (0–4)		0.040 *	
Total	2 (0–7)	2 (0–8)		0.219	

Statistical analysis: ^a^ Chi-square test; ^b^ Student *t*-test; ^c^ Mann–Whitney U test; * statistical significance, *d* Cohen’s value for effect size; NA, not applicable; PA, physical activity; MET, metabolic equivalent of task; wk, week; n, number; Q, quartile; *df*, degree of freedom for error; SD, standard deviation; VAS, Visual Analogue Scale. Statistical analysis: Student *t*-test.

**Table 3 jfmk-09-00189-t003:** Pearson correlation analyses for lumbar lordosis and trunk extensor endurance vs. pelvic tilt, trunk muscles balance (flexor/extensor ratio) and NS-LBP in female and male students.

Variable	Lordosis (°)		Trunk Extensor Endurance (sec)	
	*r*		*r*	
	Female	Male	Female	Male
Pelvic tilt (°)	0.838 ***	0.746 ***	0.152	0.290 *
Extensors/flexors ratio (balance)	0.174	0.363 *	0.625 ***	−0.007
VAS (NS-LBP)	0.003	0.157	−0.082	0.123

*r*, Pearson correlation coefficient; * *p* < 0.05 and *** *p* < 0.001; sec, seconds; VAS, Visual Analogue Scale; NS-LBP, non-specific low back pain.

**Table 4 jfmk-09-00189-t004:** Pearson correlation analyses between NS-LBP, and physical activity, time spent sitting and pelvic tilt in female and male students.

Variable	VAS (Non-Specific Low Back Pain)	
	*r*	
	Female	Male
PA (MET-min/wk)	0.253 *	−0.047
Sitting (h/day)	−0.012	−0.094
Pelvic tilt (°)	0.021	0.220
Extensors/flexors ratio (balance)	−0.166	0.094

NS-LBP, non-specific low back pain; *r*, Pearson correlation coefficient; * *p* < 0.05; VAS, Visual Analogue Scale; PA, physical activity; MET, metabolic equivalent of task; wk, week; h, hour.

## Data Availability

Data supporting this article are available from the corresponding author upon reasonable request.

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
