# Peer review of "Gender Differences in the Relationship between Physical Activity, Postural Characteristics and Non-Specific Low Back Pain in Young Adults"

_jfmk, 2024, doi:10.3390/jfmk9040189_

Round 1

Reviewer 1 Report

Comments and Suggestions for Authors

The topic of the article is worth developing. Sedentary lifestyle, improper movement habits lead to postural defects, which in turn lead to spinal pain syndromes.

The amount of repetition in the anti-plagiarism report is unacceptable. The text needs significant rewriting.

The authors examine thoracic kyphosis and lumbar lordosis as a possible deterioration of posture. In contrast, they only refer to the possible appearance of LBP, and there is no reference to thoracic pain. 

In addition, the pelvic tilt in the women studied would have to be compared to values listed as normal in anatomy.

The reviewer cannot agree with the statement that increased pelvic tilt is associated with less exercise in women. Women have a different anatomical structure of the pelvis than men, a stronger inclination of the sacrum (pelvis) and this is related to reproductive functions, As a result, the lower part of the pelvis has a different shape in both sexes. The entrance plane of the male pelvis is heart-shaped, while the female pelvis is transversely oval. The inclination of the sacrum, affects the lordosis, which is physiologically greater in women.

The increased incidence of LBP in women is rather related to their lifestyle. As for men, one would rather expect pain in the thoracic part of the back, which is due to irregularities in the form of exercises undertaken. They usually strengthen the anterior band of the body, not taking into account the extensors. It would be interesting to know whether thoracic spine pain syndrome appeared in both sexes.

The results of the study are interesting, nevertheless, according to the reviewer, the conclusions need reflection and correction.

As for the recommendations on the difference in exercise for men and women, it is clear that they must be diversified due to differences in structure, muscle strength, and hormonal and psychophysical differences.

Author Response

Reviewer 1:

Comments and Suggestions for Authors

Reviewer: The topic of the article is worth developing. Sedentary lifestyle, improper movement habits lead to postural defects, which in turn lead to spinal pain syndromes.

Response: We would like to thank the reviewers for their useful suggestions and for taking all the trouble to improve our manuscript. You can find all our answers below:

Reviewer: The amount of repetition in the anti-plagiarism report is unacceptable. The text needs significant rewriting.

Response: before submission, we always check the paper with the plagiarism detector (Turnitin) and have found the repetitions mainly in the Methods part. However, the source of these repetitions is our previous article (16% of duplication from our local repository, Faculty of Medicine, where the article is stored) (the reference is at number 32: Marijančić, V.; Grubić Kezele, T.; Peharec, S.; Dragaš-Zubalj, N.; Pavičić Žeželj, S.; Starčević-Klasan, G. Relationship between Physical Activity and Sedentary Behavior, Spinal Curvatures, Endurance and Balance of the Trunk Muscles - Extended Physical Health Analysis in Young Adults. Int. J. Environ. Res. Public Health 2023, 20, 6938. https://doi.org/10.3390/ijerph20206938).

We cannot remove or rewrite these parts of the methods as our submitted manuscript is part of the same study. We hope you understand this.

Other mostly used words are from the same paper published online in the MDPI journal Int. J. Environ. Res. Public Health (6% duplication).

And thirdly, the most frequently used words (4% duplication) come from our own paper published in the proceedings of the conference; reference at number 56, Marijančić,V.; Starčević-Klasan, G.; Grubić Kezele, T. Gender differences in physical activity and quality of life in university students of Rijeka. In Proceedings of the 2nd International Scientific Conference 4 Healthy Academic Society Poreč, Croatia, May 30-June 1, 2024. Available at: https://conference.unisport.hr/. (Accessed on 27 August 2024).

However, we have rephrased the sentences in Introduction, Discussion and partly in Limitations. We hope this is acceptable.

Reviewer: The authors examine thoracic kyphosis and lumbar lordosis as a possible deterioration of posture. In contrast, they only refer to the possible appearance of LBP, and there is no reference to thoracic pain. 

Response: we thank the reviewer for noticing this. We planned to analyze thoracic pain as well. However, we found no thoracic back pain in our pilot results. This was the main reason why we only included the analysis of LBP in this study. On the other hand, this is great idea that we will include to examine on a larger number of our students.

We have included thoracic pain and references in the introduction and limitations parts and added the explanation of why we decided to study only LBP for now (lines 49-52, lines 94-97, line 535; highlighted).

Reviewer: In addition, the pelvic tilt in the women studied would have to be compared to values listed as normal in anatomy. The reviewer cannot agree with the statement that increased pelvic tilt is associated with less exercise in women. Women have a different anatomical structure of the pelvis than men, a stronger inclination of the sacrum (pelvis) and this is related to reproductive functions, As a result, the lower part of the pelvis has a different shape in both sexes. The entrance plane of the male pelvis is heart-shaped, while the female pelvis is transversely oval. The inclination of the sacrum, affects the lordosis, which is physiologically greater in women.

Response: We are aware that women have a different anatomical structure of the pelvis than men, which we have commented on together with the pronounced lumbar lordosis. We have added the explanation that anterior pelvic tilt is a normal variant in women (line 472). In addition, we have already explained the normal range for pelvic tilt in a method section (lines 235-237), and the upper limit is 15° for both genders. The male values for upper tilt in our study (11.0±4.0) are consistent with other studies (e.g. DOI: 10.6061/clinics/2018/e647) (11.6°±5.6°). However, the female students in our study are not at the upper limit as in the aforementioned study (12.8°±6°), but have a much higher degree of anterior pelvic tilt (19.2±6.2), especially the female students with a higher levels of PA (Table 2: 21.4±5.5). In our study, this correlation is obvious. However, we have accepted to omit the statement from the text that increased pelvic tilt in women is associated with less exercise (abstract and text) to be less confusing. Accordingly, we have rephrased the main text (lines 452, 543; highlighted).

Reviewer: The increased incidence of LBP in women is rather related to their lifestyle. As for men, one would rather expect pain in the thoracic part of the back, which is due to irregularities in the form of exercises undertaken. They usually strengthen the anterior band of the body, not taking into account the extensors. It would be interesting to know whether thoracic spine pain syndrome appeared in both sexes.

Response: we agree with the reviewer that lifestyle could be the reason. Moreover, in our study we investigated the association between the type of PA and sedentary behaviour with posture and back pain. However, we did not investigate the exact type of physical activity (e.g. yoga, Pilates, which body parts are used more during housework or strength training in the gym/aerobics, etc.). We only found that women do more housework and PA at work than in their leisure time and spend more hours sitting. On the other hand, we have added some new references in line 480 (in numbers 78 and 79) to support the association between anterior pelvic tilt and low back pain.

We agree with the reviewer that thoracic spine pain should be additionally investigated. We will take this into consideration. For now, we have only analysed the LBP and included this note in the Introduction section (lines 94-97, highlighted) of the manuscript. We have added the explanation in the limitations section (line 535, highlighted).

Reviewer: The results of the study are interesting, nevertheless, according to the reviewer, the conclusions need reflection and correction. As for the recommendations on the difference in exercise for men and women, it is clear that they must be diversified due to differences in structure, muscle strength, and hormonal and psychophysical differences.

Response: we thank the reviewer very much for the good comment. We have made corrections and already pointed out the different approach to exercises for men and women in the Conclusion section (lines 553-557) and also added part of your sentence (lines 556-557, highlighted).

Reviewer 2 Report

Comments and Suggestions for Authors

The study aims to investigate gender differences in posture and low back pain (LBP) among individuals with different physical activity (PA) levels. This topic is potentially important, as the data collected could provide valuable insights into how PA affects postural and well-being in young adults.

Abstract: The current abstract is too long (>250 words) and does not have a subheading required by the journal. 

 Introduction

1. The primary objective of the study is to explore gender differences in various variables. However, the introduction lacks a sufficient explanation of the topic’s importance and does not adequately justify why the authors expect to find gender differences. The only mention of gender appears in line 62: "but also on gender and the type and intensity of sport that young people practice," no reference is provided to support this claim. This section should be significantly strengthened to justify the research aim better.

2.      Additionally, the current objective statement is unclear and, in my view, does not align well with the analyses conducted (see comments below).

Methods

3.      For sample size estimation, the authors propose r=0.25, but they do not specify what this r refers to. Since the analysis focuses on gender differences, Cohen's d would be a more appropriate effect size for estimating sample size rather than a correlation coefficient

4.      There should be some information on the reliability of spinal curvature assessments.

5.      Is your cut-off value for MET-min/wk comparable to other studies in similar populations?

6.      Since the objective is to explore gender differences in the relationship between PA and various variables, I expected some analysis to establish the relationship and compare the strength of these correlations across genders.

Results

7.      The p-value should be reported as p < 0.001 instead of p = 0.000.

8.      There is a significant imbalance in sample size for Q1, with only seven male participants in this category. This raises concerns about the validity of the t-test for comparisons in this group.

9.      Please provide effect sizes for the comparisons made.

10.  What percentage of participants is experiencing NSLBP? What is the average VAS score for those reporting LBP? This information is crucial, as analyses involving the VAS may not be valid if the majority of participants do not have NSLBP.

11.  There seems to be a discrepancy in line 448, where both genders are reported to have experienced NS-LBP with a VAS score as high as 8, but according to Table 2, no female participants reported a VAS score of 8.

Discussion

12.  Please begin the discussion with a short paragraph summarizing the key findings.
There is insufficient discussion regarding the negative correlation between posture and LBP.

13.  Line 467 could be misleading, considering that no gender differences were observed across the entire sample.

Conclusion

14.  Similar to the above comment, line 501-502 is misleading concerning the level of NSLBP in females.

15.  Based on the study, posture is not related to LBP, so the statement "However, are these postural abnormalities the exact and only problem for NS-LBP in young adults" is not valid.

Author Response

Reviewer 2:

Reviewer: The study aims to investigate gender differences in posture and low back pain (LBP) among individuals with different physical activity (PA) levels. This topic is potentially important, as the data collected could provide valuable insights into how PA affects postural and well-being in young adults. Response: We thank the reviewer for the beneficial suggestions and for taking all the effort to make our Manuscript better.

Reviewer: Abstract: The current abstract is too long (>250 words) and does not have a subheading required by the journal. 

Response: we changed the abstract to 250 words and inserted the subheadings.

 Introduction

Reviewer: 1. The primary objective of the study is to explore gender differences in various variables. However, the introduction lacks a sufficient explanation of the topic’s importance and does not adequately justify why the authors expect to find gender differences. The only mention of gender appears in line 62: "but also on gender and the type and intensity of sport that young people practice," no reference is provided to support this claim. This section should be significantly strengthened to justify the research aim better.

Response: we agree with the reviewer and have made changes according to suggestions.

Reviewer: 2.      Additionally, the current objective statement is unclear and, in my view, does not align well with the analyses conducted (see comments below).

Methods

  1. For sample size estimation, the authors propose r=0.25, but they do not specify what this r refers to. Since the analysis focuses on gender differences, Cohen's dwould be a more appropriate effect size for estimating sample size rather than a correlation coefficient

Response: we absolutely agree with the reviewer. We have adjusted this part and calculated Cohen's d to estimate the sample size and adjusted the text in the Methods - Participants accordingly (lines 105-107, highlighted).

Reviewer: 4. There should be some information on the reliability of spinal curvature assessments.

Response: we have added more explanation and new references in the methods (lines 205-208, highlighted and references at numbers 50,51,52).

Reviewer: 5.      Is your cut-off value for MET-min/wk comparable to other studies in similar populations?

Response: Usually studies do not use quartiles, as suggested in the IPAQ questionnaire, and do not mention upper values, only mean values. However, we have added a reference in the Methods (line 247 at number 55) to a similar study with university students that used quartiles (Zanovec, M.; Lakkakula, A.P.; Johnson, L.G.; Turri, G. Physical Activity is Associated with Percent Body Fat and Body Composition but not Body Mass Index in White and Black College Students. Int. J. Exerc. Sci. 2009, 15, 175–185.). The study used MET- hours/week and did not mention the upper cut-off value than only gave a value equal or above certain value as “≥”.They also did not use the IPAQ-Long Form as we did, but the IPAQ-SF. Usually, the authors use exactly the Short Form of this questionnaire, in which physical activity is assessed separately using 7 questions on vigorous and moderate physical activity in various forms, heavy lifting, aerobic exercise or fast cycling, then walking and sitting. In the Long Form of the IPAQ, physical activity at work, activity in transport, household work, housework and family care, leisure activities, sport and physical activity during leisure time, and time spent sitting are assessed separately using 27 questions.

However, we have added a further explanation in the methods (lines 247-250, highlighted) of how we included or excluded participants from the study after they had completed the questionnaire. The calculator omitted those participants who had more od MET- minutes per week above a certain value (30561) and included those who had less. The participant who had exact this value (30561 MET-min-week) had about 70 hours of MET-min per day. This value is the result of more than one physical activity per day. In fact, MET-min/week is a value that refers to the calories burned during physical activity or sitting, not the actual time spent during the activity. When calculating energy expenditure, according to the instructions for calculating the IPAQ questionnaire, all individuals who exceeded the value 30561 were marked as outliers and excluded from the study (only 2 of them).

Nevertheless, our mean values for the total MET-min/week are similar to other studies for the same population and if we calculate only the minutes/week, we also get similar values to other studies. However, there are studies like this one (Pastuszak, Anna, et al. "Level of physical activity of physical education students according to criteria of the IPAQ questionnaire and the recommendation of WHO experts" Biomedical Human Kinetics, vol. 6, no. 1, Sciendo, 2014, https://doi.org/10.2478/bhk-2014-0002) that report much higher mean values for MET-min per week for students from different universities. It should be noted that these students are studying physical education and sport and thus is expected to have much higher level of PA during a day and a week as recorded.

Reviewer: 6.      Since the objective is to explore gender differences in the relationship between PA and various variables, I expected some analysis to establish the relationship and compare the strength of these correlations across genders.

Response: we agree with the reviewer’s comment and added calculated effect sizes for these relationships (Table 1 and 2) and separated results of Pearson’s correlation for each gender (Table 3 and 4). The results are adjusted accordingly (results and table titles and legends; highlighted).

Results

Reviewer: 7.      The p-value should be reported as p < 0.001 instead of p = 0.000.

Response: we have adjusted the p value; highlighted.

Reviewer: 8.      There is a significant imbalance in sample size for Q1, with only seven male participants in this category. This raises concerns about the validity of the t-test for comparisons in this group.

  1. Please provide effect sizes for the comparisons made.

Response: we added effect sizes in Tables 1 and 2.

  1. What percentage of participants is experiencing NSLBP? What is the average VAS score for those reporting LBP? This information is crucial, as analyses involving the VAS may not be valid if the majority of participants do not have NSLBP.

Response: we have calculated the percentage of participants experiencing NS-LBP and added the results in the results part. We added the median with range for those who experiencing LBP (lines 382-383, highlighted). Lines 385-387-new results, highlighted.

Reviewer: 11.  There seems to be a discrepancy in line 448, where both genders are reported to have experienced NS-LBP with a VAS score as high as 8, but according to Table 2, no female participants reported a VAS score of 8.

Response: we have rephrased this sentence (lines 483-484, highlighted).

Discussion

Reviewer: 12.  Please begin the discussion with a short paragraph summarizing the key findings.
There is insufficient discussion regarding the negative correlation between posture and LBP.

Response: we added a short paragraph that summarize are key finding. We commented that we did not find any correlation between the LBP and posture (lines 409-415, highlighted). Now we have added more discussion (lines 487-494, lines 501-504, highlighted).

Reviewer: 13.  Line 467 could be misleading, considering that no gender differences were observed across the entire sample.

Response: we have rephrased the sentence (lines 512-516, highlighted), and have leaved “females” according to gender differences in LBP in Table 2 and new findings on LBP in Table 4.

Conclusion

Reviewer: 14.  Similar to the above comment, line 501-502 is misleading concerning the level of NSLBP in females.

Response: we have rephrased this sentence according to findings on LBP in Table 2 and Table 4. (lines 544-548, highlighted).

Reviewer: 15.  Based on the study, posture is not related to LBP, so the statement "However, are these postural abnormalities the exact and only problem for NS-LBP in young adults" is not valid.

Response: we have omitted this sentence.

Round 2

Reviewer 1 Report

Comments and Suggestions for Authors

Dear Authors,

thank you for your kind and comprehensive response.

The corrections that have been made will change the overall tone of the paper, drawing attention to the various circumstances that can affect the occurrence of back pain syndromes.

Essential differences in the shape of the spine due to gender have an impact on the appearance of sacral pain, probably also independent of physical activity, and often due to the habit of adopting incorrect posture. This indeed requires further research. It would be advisable to conduct a study within the same gender in people with bad and good posture, taking into account movement habits and lifestyle in detail. Sometimes certain exercises done incorrectly increase the chance of pain. Such results would bring researchers closer to answering the question of what influences the onset of back pain in young people.

As it stands, the article is an interesting contribution to the discussion of the topic under study and can be published as presented.  

Author Response

Reviewer 1:

Dear Authors,

thank you for your kind and comprehensive response.

The corrections that have been made will change the overall tone of the paper, drawing attention to the various circumstances that can affect the occurrence of back pain syndromes.

Essential differences in the shape of the spine due to gender have an impact on the appearance of sacral pain, probably also independent of physical activity, and often due to the habit of adopting incorrect posture. This indeed requires further research. It would be advisable to conduct a study within the same gender in people with bad and good posture, taking into account movement habits and lifestyle in detail. Sometimes certain exercises done incorrectly increase the chance of pain. Such results would bring researchers closer to answering the question of what influences the onset of back pain in young people.

As it stands, the article is an interesting contribution to the discussion of the topic under study and can be published as presented.  

Response: Dear Reviewer, thank you very much for your kind help and comments.

Reviewer 2 Report

Comments and Suggestions for Authors

I noticed that several of my comments have not been fully addressed, and explanations for omitting them were not provided. Below are the specific points that need further attention:

  1. Reliability of Spinal Curvature Assessments: Information on the reliability of these assessments is still missing. Please include this data or reference appropriate sources.

  2. Comment 8: This comment appears to have been completely ignored. 

  3. Sample Size Estimation: The authors have replaced r = 0.25 with d = 0.35 without justifying why 0.35 is a suitable effect size. How was this value determined? Is it based on previous literature? If so, please provide a reference.

  4. Newly Added Paragraph (Line 510): The statement "However, these findings should be further confirmed and investigated in more detail and on a larger group of participants" raises concerns. Are you suggesting that the current sample size is insufficient? If so, how does this align with your earlier sample size estimation? Additionally, what is the proposed larger sample size that would be considered adequate to confirm the findings?

Author Response

Reviewer 2:

I noticed that several of my comments have not been fully addressed, and explanations for omitting them were not provided. Below are the specific points that need further attention:

Reviewer: 1. Reliability of Spinal Curvature Assessments: Information on the reliability of these assessments is still missing. Please include this data or reference appropriate sources.

Response: We already inserted new references in line 211 for reliability of Spinal Mouse device on assessments of the spinal curvatures. However, we have now added an explanation and replaced some references (lines 209-216, highlighted).

Reviewer: 2. Comment 8: This comment appears to have been completely ignored.

      Response: comment 8. was: “There is a significant imbalance in sample size for Q1, with only seven male participants in this category. This raises concerns about the validity of the t-test for comparisons in this group. 9. Please provide effect sizes for the comparisons made.”

      As we categorized the participants into quartiles of PA according to different levels of MET-min/wk using the IPAQ-LF (Q1=672–2924, Q2=2925–4759, Q3=4760–7989 and Q4=7990–30561 MET-min/wk), we found only 7 males to have low PA. This low number of male students in Q1 is expected, since the male students, unlike females, are categorized as most physically active in our study. Additionally, according to your request, we provided effect sizes that are reflecting more accurately these differences. However, we included this gender comparison for Q1 group as a limitation of this study (lines 548-549, highlighted).

      Reviewer: 3. Sample Size Estimation: The authors have replaced r = 0.25 with d = 0.35 without justifying why 0.35 is a suitable effect size. How was this value determined? Is it based on previous literature? If so, please provide a reference.

      Response: We understand what you meant. We have changed this effect size into d = 0.50 and inserted the reference with the same estimation of the sample size. Accordingly, we have changed this part in the methods part and added additional explanation (lines 104-110, highlighted).

Reviewer: 4. Newly Added Paragraph (Line 510): The statement "However, these findings should be further confirmed and investigated in more detail and on a larger group of participants" raises concerns. Are you suggesting that the current sample size is insufficient? If so, how does this align with your earlier sample size estimation? Additionally, what is the proposed larger sample size that would be considered adequate to confirm the findings?

Response: We omitted this part of the sentence, not to have any misleading (line 523, highlighted).

We thank ones more Reviewers for giving their observations, suggestions and valuable comments. We appreciate it a lot and sincerely hope that the Editor and Reviewers will find this time the revision to our Manuscript satisfactory. Thank you for your consideration, we are looking forward to your answer.

Sincerely,

the authors.

Round 3

Reviewer 2 Report

Comments and Suggestions for Authors

I don't have additional comments on the manuscirpt